# Metadata Conditioning Accelerates Language Model Pre-training

**Tianyu Gao** [1]   **Alexander Wettig** [1]   **Luxi He** [1]   **Yihe Dong** [1]   **Sadhika Malladi** [1]   **Danqi Chen** [1]

## Abstract

The vast diversity of styles, domains, and quality levels present in language model pre-training corpora is essential in developing general model capabilities, but efficiently learning and deploying the correct behaviors exemplified in each of these heterogeneous data sources is challenging. To address this, we propose a new method, termed **Me**tadata **Co**nditioning then **Co**oldown (**MeCo**), to incorporate additional learning cues during pre-training. MeCo first provides metadata (e.g., URLs like `en.wikipedia.org`) alongside the text during training, then transitions to a cooldown phase using only standard text—enabling the model to perform well even without metadata. MeCo significantly accelerates pre-training across different model scales (600M to 8B parameters) and training corpora (C4, RefinedWeb, and DCLM). Notably, a 1.6B language model trained with MeCo matches the downstream task performance of standard pre-training while using 33% less data. Additionally, MeCo allows us to steer language models by conditioning the inference prompt on either real or fabricated metadata that encodes the desired output properties—for example, prepending `wikipedia.org` to reduce harmful generations or `factquizmaster.com` (fabricated) to improve performance on common knowledge tasks. We further demonstrate that MeCo is compatible with various types of metadata, such as model-generated topics. MeCo is remarkably simple, adds no computational overhead, and shows promise for producing more capable and steerable language models. Our models, data, and code are available at https://github.com/princeton-pli/MeCo.

[1]Princeton Language and Intelligence, Princeton University. Correspondence to: Tianyu Gao <tianyug@princeton.edu>.

*Proceedings of the $42^{nd}$ International Conference on Machine Learning*, Vancouver, Canada. PMLR 267, 2025. Copyright 2025 by the author(s).

## 1. Introduction

Language models (LMs) achieve remarkable general-purpose capabilities by training on vast web-sourced corpora. For instance, Internet documents about Apple CEO Tim Cook range from memes ("*Tim doesn't cook anymore*") to biographies ("*Tim Cook is the CEO of Apple*"). Treating data from these heterogeneous sources identically causes two issues: (1) it overlooks crucial contextual signals that aid comprehension, and (2) it can impede models from reliably surfacing appropriate behaviors (e.g., humor or factuality) for downstream tasks.

To provide additional information about each document's source, we propose conditioning documents with their corresponding metadata during pre-training by prepending the widely available source URLs to each document. For instance, as shown in Figure 1, adding the source URLs to Tim Cook documents helps the model distinguish among a meme, a biography, an interview article, and a financial report. To ensure the model operates effectively with or without metadata during inference, we implement a *cooldown* phase for the final 10% of training, during which we train on standard data without metadata. We call this pre-training method **Me**tadata **Co**nditioning then **Co**oldown (**MeCo**).

Metadata conditioning has been investigated in various contexts, such as steering model generations (Keskar et al., 2019), improving model robustness against malicious prompts (Korbak et al., 2023a), and enhancing knowledge memorization in synthetic settings (Allen-Zhu & Li, 2024). Distinct from prior explorations, our work establishes the general-purpose utility of this method in two crucial ways. First, we demonstrate that this paradigm can directly accelerate realistic language model pre-training and improve downstream performance. Second, the cooldown phase in MeCo ensures the model can perform inference without metadata—a key advantage over previous methods. We outline the contributions of this work below.

1. **MeCo substantially accelerates pre-training (§ 3).** We demonstrate that MeCo enables a 1.6B model to achieve the same average downstream performance as a standard pre-trained model using 33% less training data. MeCo exhibits consistent gains across model scales (600M, 1.6B, 3B, and 8B) and pre-training cor-

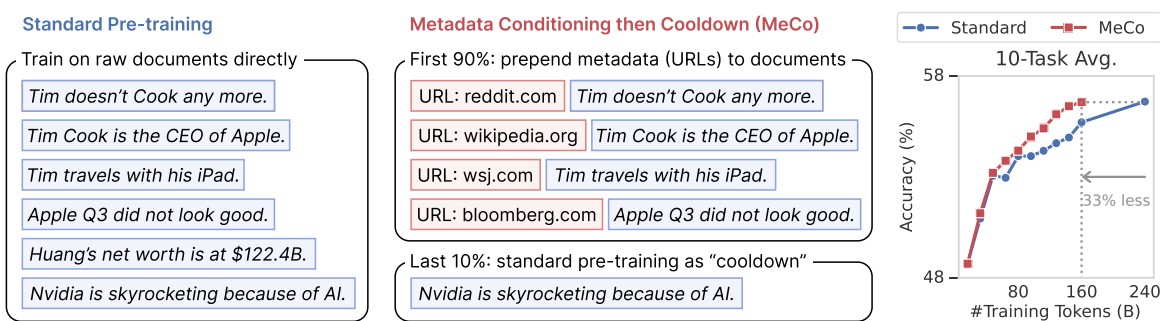

Figure 1: A comparison between data used by standard pre-training and MeCo. The figure on the right demonstrates 5-shot downstream task performance averaged across 10 tasks (1.6B models; details in §3).

pora (C4, RefinedWeb, and DCLM).

2. **MeCo unlocks a new way to steer language models (§ 4).** Prepending appropriate real or fabricated URLs to the prompt during inference can induce desired model behaviors. For example, using `factquizmaster.com` (fabricated) can enhance performance on common knowledge tasks (e.g., a 6% absolute improvement on zero-shot commonsense question answering), and using `wikipedia.org` reduces the likelihood of toxic generations several-fold compared to the standard unconditional inference.

3. **We ablate the design choices for MeCo (§5.1) and demonstrate that MeCo is compatible with various types of metadata (§ 5.2)**. Ablations using hashed URLs and model-generated topics show that metadata primarily serves to group documents by source. As such, MeCo can effectively incorporate different types of metadata, including more fine-grained ones, even when URLs are not available.

Our findings demonstrate that MeCo can significantly improve the data efficiency of language models while adding minimal computational overhead and complexity to the pre-training procedure. Moreover, the enhanced steerability afforded by MeCo holds promise in creating more controllable language models, and its general compatibility with more fine-grained and creative metadata invites further exploration. Altogether, MeCo is a simple, flexible, and effective training paradigm that simultaneously enhances both the utility and steerability of language models.

## 2. Metadata Conditioning then Cooldown

In this section, we describe our pre-training approach in detail. We assume each document in the pre-training dataset is associated with some metadata $c$. In our main experiments, we use the document URL's absolute domain name as $c$. For example, if the document's URL is `https:` `//en.wikipedia.org/wiki/Bill_Gates`, then $c$ is `en.wikipedia.org` (please refer to §5.2 for ablations on other URL variants). This URL information is readily available in many pre-training corpora, since most of them are derived from CommonCrawl[1], an open repository of web-crawled data.

Our method consists of two training stages (Figure 1):

1. **Pre-training with metadata conditioning** (first 90%): The model is trained on a concatenation of the metadata and the document, following this template: `URL:` `en.wikipedia.org\n\n[document]`. When using other types of metadata, `URL` should be replaced with the corresponding metadata name. **We only calculate the cross entropy loss over the document tokens**, disregarding those from the template or the metadata, as we found in our preliminary experiments that training on tokens from the template or the metadata slightly hurts downstream performance.

2. **Cooldown with standard data** (last 10%): Models trained solely on metadata-augmented data exhibit degraded performance when used without metadata (please refer to results in Table 4). To ensure general usage, we train the model on standard pre-training documents without any metadata during a cooldown stage, which covers the final 10% of steps in the pre-training process. The cooldown stage inherits the learning rate schedule and optimizer states from the metadata conditioning stage—i.e., it initializes the learning rate, model parameters, and optimizer states from the last checkpoint of the previous stage and continues adjusting the learning rate according to the schedule during the cooldown stage. Please refer to § A.3 for more details.

We also employ the following techniques in all our experiments, as they enhance the baseline pre-trained models' performance based on our preliminary experiments: (1) we

---

[1]`https://commoncrawl.org/`.

Table 1: Our main experimental results of pre-training a 1.6B language model on 160B tokens from DCLM. MeCo significantly outperforms standard pre-training and achieves equivalent average performance to the 240B-token baseline while using 33% less data. Interestingly, validation perplexity (PPL) does not correlate with downstream performance.

| Model | PPL | MMLU | ARC-e | ARC-c | CSQA | HSwag | OBQA | PIQA | SIQA | WG | TruQA | Avg. |
|---|---|---|---|---|---|---|---|---|---|---|---|---|
| Standard | 13.2 | 36.1 | 75.1 | 42.7 | 64.8 | 66.7 | 46.0 | 74.3 | 54.2 | 62.0 | 35.2 | 55.7 |
| *+ Data sel.* | *13.3* | *37.2* | *74.6* | *44.3* | *62.9* | *65.5* | *46.8* | *74.3* | *52.4* | *64.3* | *37.8* | *56.0* |
| *+ 80B tokens* | *12.9* | *37.1* | *75.2* | *43.2* | *64.1* | *67.7* | *49.8* | *74.7* | *54.9* | *62.8* | *37.8* | *56.7* |
| MeCo | 13.3 | 36.3 | 75.7 | 44.1 | 63.8 | 67.3 | 51.2 | 73.4 | 52.6 | 64.2 | 38.5 | **56.7** |
| | | ↑0.2 | ↑0.6 | ↑1.4 | ↓1.0 | ↑0.6 | ↑5.2 | ↓0.9 | ↓1.6 | ↑2.2 | ↑3.3 | ↑1.0 |

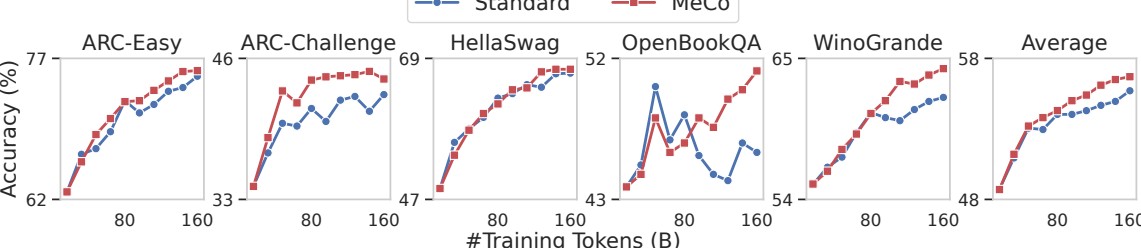

Figure 2: MeCo downstream task performance throughout training (1.6B model on DCLM). Each checkpoint of MeCo includes a 16B-token cooldown in the end. The total number of tokens used by the baseline and the corresponding MeCo checkpoints are the same for fair comparison. The reported average numbers are over all 10 tasks. Full results in Table 16.

disable cross-document attention (Dubey et al., 2024; Ding et al., 2024), which both speeds up the training (25% faster for a 1.6B model) and improves the downstream performance (§B.1); (2) when packing multiple documents into one sequence, we ensure each sequence starts with a new document rather than in the middle of one—this may result in some data being discarded when packing documents to a fixed length, but it proves beneficial for improving downstream performance.

## 3. MeCo Accelerates LM Pre-training

In this section, we demonstrate that MeCo can significantly accelerate language model pre-training (§ 3.2). We also show that MeCo leads to consistent gains across different model scales (§3.3) and training data (§3.4).

### 3.1. Experiment setup

We utilize the Llama (Touvron et al., 2023a;b; Dubey et al., 2024) version of the Transformer architecture (Vaswani et al., 2017) and the Llama-3 tokenizer for all our experiments. We conduct experiments with four different model sizes: 600M, 1.6B, 3B, and 8B. The architecture details are in §A.2. We employ standard optimization settings for language models, i.e., AdamW optimizer and cosine learning rate schedule. We follow Li et al. (2024) for hyperparameters and the details can be found in §A.1. Due to the high cost associated with pre-training and our limited resources,

we perform only one run for each experiment; however, we demonstrate in §B.2 that the variance of our experiments should be low. §A.5 outlines the computational resources required for our experiments.

**Pre-training data.** We use the best-performing open-source pre-training corpus, DCLM-Baseline (Li et al., 2024), for our main experiments. Additionally, we conduct experiments with two other data sources: a reproduction of RefinedWeb (Penedo et al., 2023) from Li et al. (2024) and the C4 dataset (Raffel et al., 2020). In the paper, we refer to these data sources as DCLM, RefinedWeb, and C4, respectively. Notably, DCLM is a subset of RefinedWeb, acquired by using a fastText classifier (Joulin et al., 2017) for selecting high-quality data (Li et al., 2024). Please refer to §A.4 for more details.

**Evaluation.** We adopt the OLMES suite (Gu et al., 2024) for evaluation, which includes the following tasks: MMLU (Hendrycks et al., 2021), ARC-Easy (ARC-e; Clark et al., 2018), ARC-Challenge (ARC-c; Clark et al., 2018), CommonsenseQA (CSQA; Talmor et al., 2019), HellaSwag (HSwag; Zellers et al., 2019), OpenBookQA (OBQA; Mihaylov et al., 2018), PIQA (Bisk et al., 2020), Social IQA (SIQA; Sap et al., 2019), and WinoGrande (WG; Sakaguchi et al., 2021). We also add the popular TruthfulQA dataset (TruQA; Lin et al., 2022). Throughout the paper, we report the average performance across all 10 tasks as "Avg.". Unless specified, we always report 5-shot in-context learning

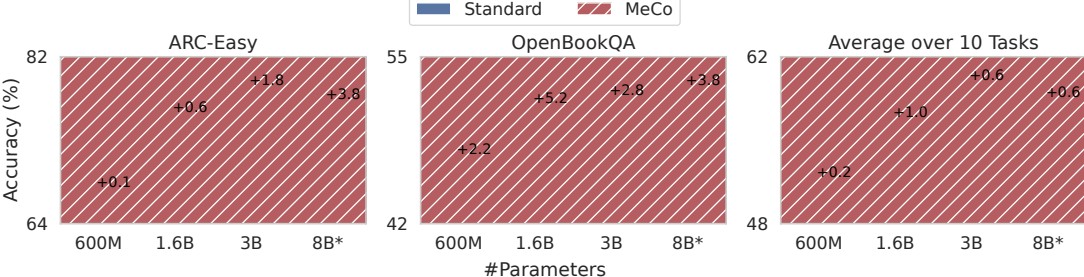

Figure 3: MeCo results across different model scales (160B tokens from DCLM except for the 8B* model, which is trained on 80B tokens due to resource constraints). Full results in Table 17. We report the average numbers across all 10 tasks. MeCo improves models across scales and leads to more gains for billion-parameter models compared to smaller models.

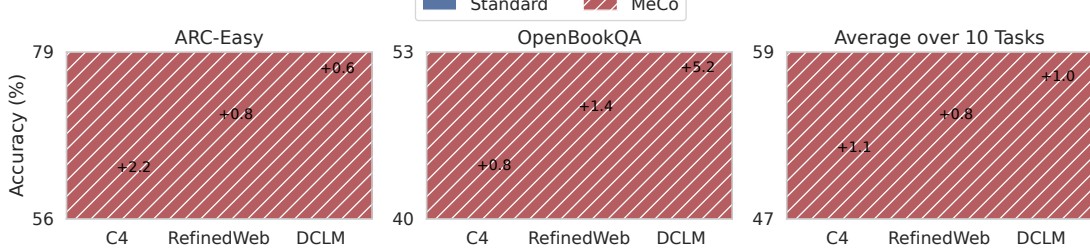

Figure 4: Results of applying MeCo over different pre-training corpora (1.6B models, 160B tokens). Full results in Table 18. We report the average numbers across all 10 tasks. MeCo provides consistent gains across different pre-training sources.

results. OLMES enhances evaluation reliability by offering three key features: (1) it provides manually-curated in-context examples for each task; (2) it evaluates with both a multiple-choice format and a cloze format, and takes the best of two; (3) it applies ablated calibration method (Brown et al., 2020; Holtzman et al., 2021) to each individual task. During evaluation, we sample 1,000 examples for each task, which improves efficiency while providing the same reliable results as full evaluation.

### 3.2. MeCo achieves comparable performance to standard pre-training with 33% less data

Table 1 shows our main results of pre-training a 1.6B language model on 160B tokens from DCLM. Besides standard pre-training (**Standard**), we also feature two other experiments, both of which use more resources and only serve as references instead of fair comparisons:

- Data selection (**+ Data sel.**): We employ the fastText data selection classifier from Li et al. (2024) to choose the top 70% documents from a 250B-token pool of DCLM data—this is similar to the high-quality data used in Section 5 of Li et al. (2024). According to the Table 4 from Li et al. (2024), this fastText classifier achieves state-of-the-art data selection performance. This method incurs additional computational cost since

the classifier must be applied over the whole corpus.

- Training with more data (**+ 80B tokens**): We train a standard model with 240B tokens, with the same optimization hyperparameters.

We first observe that MeCo achieves significantly better performance than standard pre-training across most tasks. Additionally, MeCo surpasses the data selection baseline[2]; unlike data selection, our approach does not incur any computational overhead, as it leverages readily available URL information from the pre-training data. More importantly, MeCo achieves performance comparable to standard pre-training while using 33% less data and compute, representing a substantial gain in data efficiency.

We also illustrate the changes in downstream task performance throughout the pre-training process in Figure 2. For MeCo, each checkpoint in the figure includes a cooldown phase on 16B tokens (10% of the total training tokens). For instance, the 80B checkpoint consists of 64B tokens of conditional training followed by 16B tokens of cooldown.

---

[2]It is important to note that the DCLM data is already a subset of the RefinedWeb data selected by this classifier. We do not claim that MeCo consistently outperforms data selection; rather, we demonstrate that MeCo can be integrated with data selection to achieve further improvements, while data selection alone tends to yield diminishing returns.

Table 2: Conditional inference further improves MeCo performance (full results in Table 19).

| Inference | Pre-training | |
| --- | --- | --- |
| | Standard | MeCo |
| Unconditional | 55.7 | 56.7 |
| Conditional | 55.8 ↑0.1 | 57.2 ↑0.5 |

We observe that MeCo consistently surpasses the baseline model, particularly in the later stage of training.

**Discussion of perplexity.** Table 1 reveals that validation perplexity does not correlate with downstream performance in our experiments. Notably, when comparing the 240B baseline to the 160B MeCo model, the baseline exhibits much lower perplexity due to the larger data size, yet the two models achieve similar average downstream performance. This observation aligns with previous studies (Tay et al., 2022; Liu et al., 2023; Wettig et al., 2024) indicating that perplexity is not always a reliable indicator of downstream performance; the final task performance can be impacted by other critical factors, such as inductive bias.

### 3.3. MeCo improves performance across model scales

Figure 3 demonstrates the results across different model scales (600M, 1.6B, 3B, and 8B). We train all the models with the same optimization hyperparameters and the same amount of data (160B on DCLM) except for the 8B model, which is trained on 80B tokens with a lower learning rate due to resource constraints and training instability (details in §A.1).

We first observe that MeCo improves model performance across all scales. Although the trend is somewhat noisy, MeCo appears to yield greater improvements for larger models, with billion-parameter models showing more significant gains compared to the 600M model. Note that this is a qualitative observation, as downstream task performance is known to scale less smoothly compared to pre-training loss.

### 3.4. MeCo improves performance across different training corpora

We train 1.6B models on 160B tokens from three different data sources: C4, RefinedWeb, and DCLM. We present the results in Figure 4. If we use the average downstream performance as an indicator for data quality, we can rank the three data sources as DCLM > RefinedWeb > C4. We observe that MeCo provides consistent and significant gains across different data sources, both on the average accuracies and individual tasks.

## 4. Conditional Inference Steers Language Model Generations

MeCo not only improves the general quality of pre-trained language models (evaluated by standard few-shot downstream task performance), but also unlocks the possibility of steering the model's generations during inference by conditioning it on particular URLs. We term this paradigm **conditional inference**, as illustrated in Figure 5.

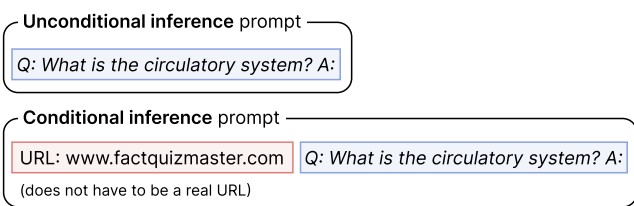

Figure 5: Illustration of conditional inference: We can condition the model by prepending a URL to the prompt. The URL does not need to be a real one.

Steering language model generations by conditioning the model on a "control sequence" has been explored in the past, either for style control (Keskar et al., 2019) or for avoiding harmful content (Korbak et al., 2023a). In this section, we study how combining conditional inference and MeCo (even with cooldown) can both improve the downstream task performance and reduce the likelihood of harmful generations.

### 4.1. Conditional inference improves MeCo's downstream task performance

In this section, we demonstrate how prepending appropriate URLs to the inputs improves MeCo's downstream performance. We first design a URL for each downstream task used in our evaluation, for example, `www.factquizmaster.com` for OpenBookQA and `www.socialskillsassessment.com` for Social IQA. You can find all the customized URLs in Table 11. We note that (1) the URLs do not need to be real; (2) we did not use trial-and-error when choosing the URLs to avoid overfitting to the test set.

We apply the same set of customized URLs to both the standard model and MeCo (1.6B, 160B DCLM tokens) and the results are shown in Table 2. We see that applying conditional inference leads to little difference on the standard model but a significant improvement on MeCo. Overall, MeCo with conditional inference achieves 1.5% absolute improvement compared to standard pre-training with unconditional inference.

We also explore the impact of different URLs on performance, as shown in Table 3. In this experiment, we use two real URLs: `boards.4chan.org`, an anonymous imageboard known for its association with offensive content,

Table 3: Zero-shot evaluation of MeCo (1.6B, 160B DCLM tokens) with different URLs. We show the delta between unconditional inference and using URLs.

| Inference URLs | ARC-e | ARC-c | CSQA | OBQA |
|---|---|---|---|---|
| Unconditional | 69.6 | 43.2 | 54.7 | 48.4 |
| 4chan.org | 66.7 ↓2.9 | 41.1 ↓2.1 | 53.6 ↓1.1 | 47.8 ↓0.6 |
| factmonster.com | 70.7 ↑1.1 | 45.7 ↑2.5 | 60.9 ↑6.2 | 52.4 ↑4.0 |

and www.factmonster.com, a trivia website. Unlike our main experiment, we employ *zero-shot* prompting to highlight the effects of different URLs. Our findings indicate that selecting an appropriate URL can significantly enhance zero-shot results compared to using a more adversarial one: for example, using factmonster.com outperforms 4chan.org by 7.3% on CommonsenseQA.

### 4.2. MeCo with conditional inference reduces harmful generations from the model

In addition to improving downstream task performance, MeCo with conditional inference also reduces harmful generations. To evaluate the toxicity of model generations, we follow (Korbak et al., 2023b) to sample 4096 text sequences from the models, with temperature $T = 0.7$ and top-$p$=0.9. The generated sequences have lengths between 10 and 128 tokens. For unconditional inference, the model is only conditioned on the BOS token. For conditional inference, the model is conditioned on en.wikipedia.org.

To obtain toxicity scores, we follow the setup in (Korbak et al., 2023b) and use the toxic comment classifier Detoxify (Hanu & Unitary team, 2020). We use the unbiased model from Detoxify, which is based on RoBERTa (Liu et al., 2019) and trained on a human-labeled dataset of nearly 2 million comments, created for the task of evaluating unintended bias (Borkan et al., 2019). The classifier provides both general toxicity scores and more granular scores (e.g., obscene, insult).

We show the averaged toxicity scores over all sampled generations in Figure 6. We observe that using en.wikipedia.org for conditional inference reduces the toxicity scores of generations from both the standard pretraining model and MeCo. Conditional inference is more effective on MeCo, leading to a significantly lower toxicity score compared to the baseline.

## 5. Ablation Studies

### 5.1. Different strategies for mixing metadata-conditioned and standard data

In this section, we study the best strategy to mix metadata-augmented data and standard data. We experiment with

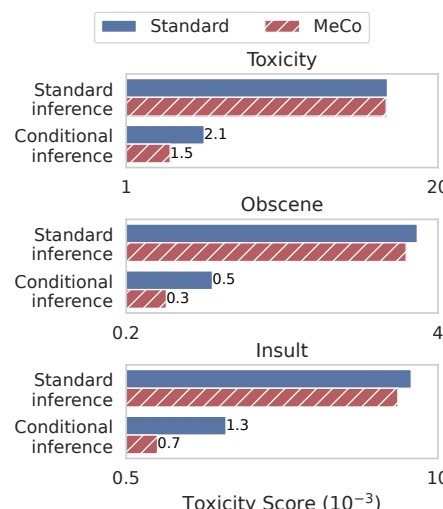

Figure 6: MeCo with conditional inference (using en.wikipedia.org) significantly reduces harmful generations from the model.

four different strategies: only standard data, only metadata-conditioned data, directly mixing the two sources of data throughout training (90% URL + 10% standard) and two-stage training (i.e., first 90% with metadata conditioning and then 10% standard data)—the last one is MeCo.

Table 4 demonstrates the results of the different mixing strategies. First, we see that only training on metadata-conditioned data leads to performance degradation, emphasizing the importance of cooldown. While both directly mixing the two types of data and two-stage training improve the performance compared to the standard pre-training baseline, first training on metadata-conditioned data and then cooldown with standard data leads to better and more consistent gains. We also perform additional ablations on the length of cooldown in § B.3, which show that 10%-20% cooldown achieves the best performance (and we use 10% in our experiments).

### 5.2. Understanding the role of metadata

To better understand how MeCo works, we experiment with various types of metadata and present the results in Table 5. Below, we describe these metadata types and their outcomes.

**URL variants.** We test URL variants that provide more information (full URLs) and less information (URL suffixes). While full URLs perform similarly to MeCo, using URL suffixes results in significant performance degradation, suggesting that absolute domain names (e.g., en.wikipedia.org) provide the appropriate granularity as metadata.

Table 4: Different strategies of mixing metadata-augmented and standard data. Full results in Table 20.

| Model | ARC-e | ARC-c | HSwag | OBQA | 10-Task Avg. |
|---|---|---|---|---|---|
| 100% standard | 75.1 | 42.7 | 66.7 | 46.0 | 55.7 |
| 100% URL | 72.4 ↓2.7 | 28.8 ↓13.9 | 61.5 ↓5.2 | 42.6 ↓3.4 | 50.3 ↓5.4 |
| 90% URL + 10% standard | 72.5 ↓2.6 | 43.1 ↑0.4 | 66.9 ↑0.2 | 50.0 ↑4.0 | 56.4 ↑0.7 |
| MeCo | **75.7** ↑0.6 | **44.1** ↑1.4 | **67.3** ↑0.6 | **51.2** ↑5.2 | **56.7** ↑1.0 |

Table 5: Ablations on using different metadata for MeCo. The average results are over all 10 tasks. Full results in Table 21.

| Metadata | Examples | Avg. |
|---|---|---|
| URLs (MeCo) | `en.wikipedia.org` | 56.7 |
| Full URLs | `en.wikipedia.org/wiki/Bill_Gates` | 56.8 ↑0.1 |
| URL suffixes | `org` | 56.2 ↓0.6 |
| Top 0.2% URLs (covering 42% texts) | `en.wikipedia.org` or `unknown` | 56.4 ↓0.3 |
| Top 2% URLs (covering 65% texts) | `en.wikipedia.org` or `unknown` | 56.3 ↓0.4 |
| Hashed URLs | `7dsjuj3a-olp0` | 56.7 ↑0.0 |
| Model-generated topics | `Technology leader biography` | 56.6 ↓0.1 |

**Top URLs.** We retain only the most frequently appearing URLs from the DCLM data and mark others as "unknown". We experiment with two tiers: top 0.2% URLs (each URL corresponds to roughly more than 1,000 documents, covering 41.6% of the DCLM data) and top 2% URLs (each URL corresponds to more than 100 documents, covering 65.1% of the DCLM data). The URL distribution in DCLM is highly skewed, with a few top URLs covering a large portion of the data. Examples of top URLs are shown in Table 15. This experiment aims to determine whether MeCo primarily benefits from modeling infrequent or high-frequency URLs. We find that using only top URLs does not match MeCo's performance, indicating that MeCo also benefits from low-frequency URLs.

**Hashed URLs.** We map each unique URL into a random string to investigate whether MeCo needs to learn the semantics of URLs or simply recognizes that certain documents belong to the same groups. Surprisingly, using hashed URLs achieves performance on par with semantically-meaningful URLs, indicating that the semantic meaning of the metadata is not necessary for better pre-trained models—instead, simply providing signals that group certain documents together is sufficient for improving pre-training data efficiency.

**Model-generated topics.** We explore ways of generating metadata in case readily available metadata is absent or insufficient. We prompt a Llama-3.1-8B-Instruct model to generate a two-word or three-word topic for each document, such as "technology leader biography" or "gaming forum" (more details in §A.6). This is more fine-grained metadata compared to domains (e.g., "Wikipedia" or "Books"). Note that prompting models to generate topics is extremely ex-

pensive, taking roughly 1,500 GPU hours, similar to what is required to pre-train the 1.6B model. Hence, it is not a practical method but included for analysis purposes. We observe that using model-generated topics leads to similar results to our main MeCo model, suggesting that metadata based on document contents instead of sources is equally useful, prompting future explorations on more creative ways of generating metadata.

Our ablations suggest that metadata conditioning improves pre-training data efficiency by grouping documents together by source or topic. We propose two preliminary hypotheses as to how metadata conditioning affects model training: First, the model may automatically learn to prioritize documents from useful sources or topics, thereby internally optimizing the mixture of training domains, which has been shown to be useful during pre-training (Xie et al., 2024; Jiang et al., 2024). Indeed, Allen-Zhu & Li (2024) also suggested that language models may autonomously identify domains rich in knowledge. Second, the model may use the additional metadata supervision to simply learn more structured representations of these large corpora, with no knowledge of the quality of each of the groups. We believe that the precise mechanism by which MeCo accelerates pre-training and improves model steerability warrants further theoretical and empirical study.

## 6. Related Work

**Metadata conditioning.** CTRL (Keskar et al., 2019) first proposed "conditional language models" for controlled generation: the method prepended the pre-training documents with "control codes" such as source domains, which allowed

for steering the generation during inference by prompting the model with different control codes. Dhingra et al. (2022) used timestamps as the metadata to train time-aware language models and Liu et al. (2020) adopted document languages as the metadata for a multilingual pre-trained model. Aghajanyan et al. (2022) pre-trained language models on hyper text, which provided extra metadata such as class and id, which allowed for conditional inference as well. Kyrylov & Chaplynskyi (2023) pre-trained language models on Ukrainian text conditioned on metadata. Weller et al. (2024) demonstrated that prompting models with text like "according to Wikipedia" improves their performance. Conditional training was also explored in alignment and preference optimization: Korbak et al. (2023a) pre-trained models with reward model scores as the prefix and Lu et al. (2022); Liu et al. (2024) conditioned the text on their quality measurements in post-training—both allowed prompting the model with a high quality score during inference to output more human-preferred text. Besides, Khalifa et al. (2024) used a similar idea to inject "document IDs" into the pre-training corpus to enable training data attribution, though the "IDs" were appended, instead of prepended.

Recently, Allen-Zhu & Li (2024) investigated language models' ability to memorize knowledge by using synthetically generated biographical data. They trained models on a mixture of such data and unrelated data, and tested models on recalling the biographical information. They found that prepending a special token to the biographical data enhanced the model's memorization capacity. The authors argued that this technique helped models recognize high-quality sources and was analogous to adding URLs to pre-training documents. However, the controlled setting in Allen-Zhu & Li (2024) was limited to two synthetic data sources and did not incorporate real URLs, making it fundamentally different from our experimental setup and contributions.

We also highlight two concurrent works: Zhu et al. (2025) and Wang et al. (2025). The former uses synthetic experiments and theoretical analysis to show that context-enhanced learning—such as prepending metadata—can improve sample efficiency. The latter demonstrates the benefits of conditional generative modeling when source distributions share certain similarities.

While our idea and findings echo previous and concurrent literature, our paper is the first to explore the use of metadata conditioning in modern-scale LM pre-training and its effect on downstream task performance. Compared to other types of metadata explored by prior work, we use URLs as they can be acquired with no additional cost and they are more informative than source domains or reward scores.

**Selecting pre-training data.** The quality of pre-training corpora is essential for the performance of the resulting language models. Consequently, there has been a huge amount of effort invested into improving pre-training data, starting from heuristic-based filtering (Raffel et al., 2020; Rae et al., 2021; Laurençon et al., 2022; Penedo et al., 2023; Soldaini et al., 2024) and deduplication (Lee et al., 2022; Anil et al., 2023; Touvron et al., 2023a; Abbas et al., 2023). Recently, model-based data filtering or data selection has emerged: many works sought to use simple ngram models to select those that resemble high-quality domains such as Wikipedia (Brown et al., 2020; Xie et al., 2023; Li et al., 2024) or to use an existing language model for perplexity filtering (Wenzek et al., 2020; Muennighoff et al., 2023; Marion et al., 2023). Gunasekar et al. (2023); Wettig et al. (2024); Penedo et al. (2024); Dubey et al. (2024) instead used a large language model to score instances based on abstract values such as whether they are "educational"—but these methods introduce considerable overheads as running these language models over the whole pre-training corpus is costly and whether they can lead to better performance under the same computational budget is unclear (Goyal et al., 2024; Kaddour et al., 2024).

Another line of works aimed to adjust the domain mixture for more data-efficient training (Xie et al., 2024; Xia et al., 2024; Jiang et al., 2024). However, these models require an existing domain taxonomy (which is usually very coarse-grained) and a target loss to optimize for—which has been shown to not always correlate with downstream performance (Tay et al., 2022; Liu et al., 2023).

Recently, Wettig et al. (2025) introduced a method for constructing domain taxonomies and automatically annotating pre-training data—with the domain annotations, they further explored optimizing domain mixtures for better downstream performance. This approach also highlighted the link between two data selection approaches mentioned before: applying quality filtering implicitly changes the data domain mixture. We find a connection to our work as well: while Wettig et al. (2025) focus on annotating data with coarse-grained domains, we utilize URLs to provide more fine-grained domain information.

## 7. Conclusion

We introduce metadata conditioning then cooldown (MeCo), an extremely simple method that consistently outperforms standard pre-training while incurring negligible computational overhead. MeCo leverages commonly available metadata, such as source URLs, by prepending them to pre-training documents. At the end of training, MeCo removes the URLs from the data to enable inference without metadata. Through comprehensive experiments across various model scales and training corpora, we demonstrate MeCo's effectiveness, achieving up to a 33% speedup in pre-training. Additionally, we show that prompting MeCo models with

suitable metadata can further enhance their downstream performance and mitigate harmful outputs. Our findings underscore the potential of metadata conditioning to enhance data efficiency in pre-training and to develop more controllable and steerable language models.

## Impact Statement

MeCo is a simple, general method that can significantly accelerate language model pre-training, making it highly applicable in both industrial applications and academic research. MeCo also enables steering language models to output less harmful content, potentially supporting LM safety research and the safe deployment of LMs in real-world applications.

## Limitations

Due to limited resources and the costly nature of pre-training, we do not perform multi-run experiments; however, we show in §B.2 that the variance of our experiments should be low and our results are significant. All our investigations are limited to English corpora. We do not study the interplay between metadata conditioning and post-training procedures. We also do not have a mechanistic understanding of how conditioning on metadata helps improve the downstream performance. We hope our results can shed light on these interesting questions and motivate further research on metadata conditioning.

## Acknowledgments

We acknowledge Angelica Chen, Sanjeev Arora, Kyunghyun Cho, Yisong Yue, Luca Soldaini, and members of Princeton Language and Intelligence for their helpful feedback and discussion. Tianyu Gao is supported by an IBM PhD Fellowship. This research is funded by the National Science Foundation (IIS-2211779) and a Sloan Research Fellowship.

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

# A. Experiment Details

## A.1. Hyperparameters

Table 6 shows the hyperparameter settings used in our experiments. We follow Li et al. (2024) for the high learning rate and weight decay except for the 8B model, which requires a lower learning rate for numerical stability.

| Hyperparameters | Values |
|---|---|
| Optimizer | AdamW ($\beta_1 = 0.9$, $\beta_2 = 0.95$) |
| Learning rate | $3e-3$ ($5e-4$ for the 8B model) |
| Weight decay | 0.033 (0.1 for the 8B model) |
| Batch size | 4M tokens |
| Warmup | 5% linear warmup |
| Schedule | Cosine decay to 10% of the peak learning rate |
| Seq length | Pack to 8192 tokens |

Table 6: Hyperparameter settings for our experiments.

## A.2. Model configurations

We use the Llama variant (Touvron et al., 2023a) of Transformers (Vaswani et al., 2017) for our experiments. All models use the Llama-3 tokenizer (Dubey et al., 2024). We add a BOS and an EOS token at the beginning and end of every document. The detailed configurations are specified in Table 7.

| #Param | #Layers | Hidden | Intermediate | #Heads | Head Dim |
|---|---|---|---|---|---|
| 600M | 24 | 1024 | 4096 | 16 | 64 |
| 1.6B | 24 | 2048 | 5504 | 16 | 128 |
| 3B | 28 | 3072 | 8192 | 24 | 128 |
| 8B | 32 | 4096 | 14336 | 32 | 128 |

Table 7: Model configurations for our experiments.

## A.3. Cooldown details

The metadata conditioning stage (90%) and the cooldown stage (10%) share the same learning rate schedule—i.e., the metadata conditioning stage will end at the 90% of the learning rate schedule and the cooldown stage will resume from that same point on the schedule and continue the learning rate decay. It also inherits all the optimizer states. To ensure the cooldown stage does not see repeated data as the conditional training stage, we use a different subset of data for cooldown for all our DCLM experiments.

For our 8B experiments (80B tokens), due to the checkpoint saving configuration, we performed a 10B-token cooldown (12.5% instead of 10% of the total training).

## A.4. Dataset details

Table 8 shows the dataset details for our pre-training experiments.

| Dataset | Description |
|---|---|
| C4 | The SlimPajama (Soboleva et al., 2023) C4 subset |
| RefinedWeb | DCLM-reproduced (Li et al., 2024) RefinedWeb |
| DCLM | DCLM-Baseline, which is a filtered version of DCLM-reproduced RefinedWeb |

Table 8: Pre-training dataset details.

### A.5. Experimental resource

Table 9 shows the resources required to train the models in our experiments. Our main models (1.6B, 160B tokens) take roughly 2 days to train on 32 H100 GPUs.

| #Params | 600M | 1.6B | 1.6B | 3B | 8B |
|---|---|---|---|---|---|
| #Tokens | 160B | 160B | 240B | 160B | 80B |
| #GPU hours | 776 | 1536 | 2304 | 3085 | 3905 |

Table 9: Resources required to train the models in our experiments (H100 GPU hours).

### A.6. Prompts for model-generated topics

Table 10 shows the prompt used for generating topics. We prompt a Llama-3.1-8B-Instruct model to generate topics. We only use the first 1024 tokens from the document as the snippet. We use greedy decoding.

```
Based on the given sampled snippet from a document (could be a webpage, a book, a
codebase, a paper, or anything else), write a domain keyphrase (within 4 words; for
example, code, international news, food blog, biography, science fiction, politics essay,
gaming forum, algebra quiz, physics textbook, restaurant advertisement, religous story,
etc.)  for the document.  The "domain keyphrase" should consider both the topics and the
genre/source of the document.

*** Start of the snippet ***

{{snippet}}

*** End of the snippet ***

Now output the domain (do not output other things):
```

Table 10: The prompt for generating topics.

### A.7. Customized URLs for conditional inference

Table 11 shows the customized URLs for conditional inference.

| Tasks | Customized URLs |
|---|---|
| MMLU | www.testprepportal.com |
| ARC-Easy | www.sciencestudyquiz.com |
| ARC-Challenge | www.sciencestudyquiz.com |
| CommonsenseQA | www.quizsmart.com |
| HellaSwag | www.wikihowquiz.com |
| OpenBookQA | www.factquizmaster.com |
| PIQA | www.basicknowledgequiz.com |
| Social IQA | www.socialskillsassessment.com |
| WinoGrande | www.testpreppractice.com |
| TruthfulQA | www.factcheckfun.com |

Table 11: Customized URLs for conditional inference.

## B. Additional Experiments

### B.1. Cross-document attention ablation

Table 12 shows a comparison between enabling and disabling cross-document attention. Disabling cross-document attention leads to significant speedups for our training (for a 1.6B model, it is 25% faster). We also see that it brings a considerable

performance improvement on the vanilla model. Interestingly, the average performance does not differ much between two different attention patterns for MeCo, suggesting that prepending the URLs to the document helps the model learn the noisy cross-document attention. Based on these results, all other experiments in this paper disable cross-document attention.

| Model | MMLU | ARC-e | ARC-c | CSQA | HSwag | OBQA | PIQA | SIQA | WG | TruQA | Avg. |
|---|---|---|---|---|---|---|---|---|---|---|---|
| Standard | 36.1 | 75.1 | 42.7 | 64.8 | 66.7 | 46.0 | 74.3 | 54.2 | 62.0 | 35.2 | 55.7 |
| +Cross-doc attn | 36.3 | 73.4 | 41.6 | 63.2 | 65.5 | 46.0 | 73.6 | 52.4 | 61.3 | 36.7 | 55.0 |
| MeCo | 36.3 | 75.7 | 44.1 | 63.8 | 67.3 | 51.2 | 73.4 | 52.6 | 64.2 | 38.5 | 56.7 |
| +Cross-doc attn | 35.5 | 72.7 | 45.4 | 66.3 | 66.1 | 51.8 | 74.4 | 52.8 | 62.4 | 38.2 | 56.6 |

Table 12: Cross-document attention ablation (160B tokens, 1.6B parameters).

## B.2. Experiment variance

Due to the nature of pre-training experiments and the high cost associated with it, we perform single runs for all our experiments and do not report their standard deviations. However, we provide a reference point here for estimating the variance of our experiments. We take the 90% checkpoint of the 1.6B-parameter, 160B-token standard pre-training model, and continue the rest 10% of the training with three disjoint sets of data. Table 13 shows their performance. We see that while some individual tasks show performance differences, the standard deviation of the average performance is very low (0.1%), demonstrating that the average performance across our selected tasks is an indicative and stable metric.

| Model | MMLU | ARC-e | ARC-c | CSQA | HSwag | OBQA | PIQA | SIQA | WG | TruQA | Avg. |
|---|---|---|---|---|---|---|---|---|---|---|---|
| Standard run 1 | 36.1 | 75.1 | 42.7 | 64.8 | 66.7 | 46.0 | 74.3 | 54.2 | 62.0 | 35.2 | 55.7 |
| Standard run 2 | 36.2 | 73.9 | 43.4 | 63.1 | 67.5 | 46.2 | 74.2 | 53.2 | 62.0 | 35.5 | 55.5 |
| Standard run 3 | 36.3 | 73.8 | 43.2 | 63.4 | 67.5 | 45.8 | 74.5 | 54.2 | 62.8 | 34.7 | 55.6 |
| Avg. | 36.2 | 74.3 | 43.1 | 63.8 | 67.2 | 46.0 | 74.3 | 53.9 | 62.3 | 35.1 | 55.6 |
| Std. | ±0.1 | ±0.7 | ±0.4 | ±0.9 | ±0.5 | ±0.2 | ±0.2 | ±0.6 | ±0.5 | ±0.4 | ±0.1 |

Table 13: Multiple runs of the baseline model (1.6B parameters, 160B tokens from DCLM). The average performance across runs shows low variance.

## B.3. Cooldown length ablation

Table 14 shows the performance of different cooldown lengths. We see that performing a 10% and 20% cooldown achieves similar results, while further increasing the length hurts the performance. For simplicity, we use 10% cooldown for all our experiments. We note that the best cooldown length can vary across different numbers of parameters, total numbers of training tokens, and the pre-training corpora; however, performing such a fine-grained search across all different settings is intractable.

| Model | MMLU | ARC-e | ARC-c | CSQA | HSwag | OBQA | PIQA | SIQA | WG | TruQA | Avg. |
|---|---|---|---|---|---|---|---|---|---|---|---|
| 10% cooldown | 36.3 | 75.7 | 44.1 | 63.8 | 67.3 | 51.2 | 73.4 | 52.6 | 64.2 | 38.5 | 56.7 |
| 20% cooldown | 36.5 | 74.7 | 46.0 | 64.2 | 67.1 | 49.4 | 73.6 | 53.3 | 64.3 | 39.0 | 56.8 |
| 30% cooldown | 36.7 | 74.8 | 45.0 | 60.9 | 67.5 | 49.0 | 74.2 | 51.6 | 62.8 | 39.2 | 56.2 |

Table 14: Ablations on different cooldown lengths (1.6B parameters, 160B tokens).

## C. DCLM URL Distributions

Table 15 shows the top 50 URLs from DCLM and the corresponding document ratios.

## D. Full Results

Table 16, Table 18, Table 17, Table 20, Table 21, and Table 19 show the detailed results of experiments reported in our main paper.

| URLs | Document ratios |
|------|-----------------|
| en.wikipedia.org | 0.256% |
| stackoverflow.com | 0.240% |
| www.theguardian.com | 0.207% |
| www.urbandictionary.com | 0.149% |
| www.fanfiction.net | 0.148% |
| www.businessinsider.com | 0.139% |
| gizmodo.com | 0.123% |
| everything2.com | 0.119% |
| www.physicsforums.com | 0.100% |
| www.reference.com | 0.090% |
| www.theatlantic.com | 0.087% |
| www.mumsnet.com | 0.086% |
| superuser.com | 0.086% |
| chowhound.chow.com | 0.085% |
| www.huffingtonpost.com | 0.082% |
| serverfault.com | 0.082% |
| www.engadget.com | 0.079% |
| math.stackexchange.com | 0.078% |
| www.nytimes.com | 0.075% |
| news.bbc.co.uk | 0.073% |
| gawker.com | 0.071% |
| tvtropes.org | 0.069% |
| www.instructables.com | 0.069% |
| www.fool.com | 0.068% |
| www.enotes.com | 0.067% |
| townhall.com | 0.067% |
| slashdot.org | 0.066% |
| www.foxnews.com | 0.066% |
| kotaku.com | 0.066% |
| articles.chicagotribune.com | 0.064% |
| www.reddit.com | 0.063% |
| www.complex.com | 0.063% |
| jezebel.com | 0.062% |
| www.gamefaqs.com | 0.061% |
| www.aljazeera.com | 0.061% |
| askubuntu.com | 0.061% |
| abcnews.go.com | 0.060% |
| mathoverflow.net | 0.058% |
| www.csmonitor.com | 0.058% |
| articles.latimes.com | 0.058% |
| www.bookrags.com | 0.057% |
| lifehacker.com | 0.057% |
| www.sfgate.com | 0.057% |
| jalopnik.com | 0.057% |
| www.ancestry.com | 0.057% |
| www.nifty.org | 0.057% |
| www.theregister.co.uk | 0.057% |
| www.osnews.com | 0.056% |
| www.cnet.com | 0.055% |
| www.ign.com | 0.055% |

Table 15: Top 50 URLs from DCLM.

| #Tokens | MMLU | ARC-e | ARC-c | CSQA | HSwag | OBQA | PIQA | SIQA | WG | TruQA | Avg. |
|---|---|---|---|---|---|---|---|---|---|---|---|
| | | | | | Standard | | | | | | |
| 16B | 30.4 | 62.8 | 34.2 | 56.0 | 48.7 | 43.8 | 69.9 | 47.2 | 55.2 | 39.1 | 48.7 |
| 32B | 32.1 | 66.8 | 37.3 | 60.0 | 55.9 | 45.2 | 70.3 | 46.7 | 56.5 | 38.6 | 50.9 |
| 48B | 34.1 | 67.4 | 40.0 | 60.9 | 58.0 | 50.2 | 71.8 | 52.5 | 57.3 | 38.3 | 53.1 |
| 64B | 34.0 | 69.2 | 39.8 | 61.6 | 59.8 | 46.8 | 72.7 | 50.2 | 59.2 | 36.3 | 53.0 |
| 80B | 34.9 | 72.5 | 41.4 | 58.6 | 62.8 | 48.4 | 72.8 | 52.7 | 60.8 | 35.5 | 54.0 |
| 96B | 34.9 | 71.2 | 40.2 | 62.1 | 63.5 | 45.8 | 72.4 | 53.5 | 60.4 | 36.4 | 54.0 |
| 112B | 35.6 | 72.1 | 42.2 | 62.9 | 64.9 | 44.6 | 73.3 | 52.6 | 60.1 | 34.6 | 54.3 |
| 128B | 35.9 | 73.5 | 42.5 | 62.8 | 64.5 | 44.2 | 73.1 | 53.9 | 61.0 | 35.3 | 54.7 |
| 144B | 36.1 | 73.9 | 41.1 | 60.6 | 66.6 | 46.6 | 73.5 | 53.9 | 61.6 | 35.5 | 55.0 |
| 160B | 36.1 | 75.1 | 42.7 | 64.8 | 66.7 | 46.0 | 74.3 | 54.2 | 62.0 | 35.2 | 55.7 |
| | | | | | MeCo | | | | | | |
| 16B | 30.4 | 62.8 | 34.2 | 56.0 | 48.7 | 43.8 | 69.9 | 47.2 | 55.2 | 39.1 | 48.7 |
| 32B | 32.5 | 66.0 | 38.7 | 58.2 | 53.9 | 44.6 | 70.6 | 49.4 | 56.2 | 41.8 | 51.2 |
| 48B | 34.0 | 68.9 | 43.0 | 59.2 | 57.8 | 48.2 | 71.6 | 50.4 | 57.9 | 41.2 | 53.2 |
| 64B | 34.2 | 70.6 | 41.9 | 62.6 | 60.4 | 46.0 | 72.1 | 50.5 | 59.1 | 40.1 | 53.8 |
| 80B | 34.3 | 72.4 | 44.0 | 61.7 | 61.9 | 46.6 | 72.6 | 49.4 | 60.7 | 39.1 | 54.3 |
| 96B | 34.9 | 72.5 | 44.3 | 63.1 | 64.1 | 48.2 | 72.9 | 49.5 | 61.7 | 38.7 | 55.0 |
| 112B | 35.4 | 73.6 | 44.4 | 63.6 | 64.4 | 47.6 | 72.4 | 51.4 | 63.2 | 37.8 | 55.4 |
| 128B | 35.7 | 74.6 | 44.5 | 64.9 | 66.9 | 49.4 | 73.0 | 51.5 | 63.0 | 37.5 | 56.1 |
| 144B | 36.1 | 75.6 | 44.8 | 63.6 | 67.3 | 50.0 | 73.8 | 52.1 | 63.7 | 38.0 | 56.5 |
| 160B | 36.3 | 75.7 | 44.1 | 63.8 | 67.3 | 51.2 | 73.4 | 52.6 | 64.2 | 38.5 | 56.7 |

Table 16: Intermediate checkpoint results for the 1.6B-parameter, 160B-token runs. For all MeCo checkpoints, we perform a 16B-token cooldown (i.e., the 64B checkpoint is 48B metadata conditioning training + 16B cooldown).

| Model | MMLU | ARC-e | ARC-c | CSQA | HSwag | OBQA | PIQA | SIQA | WG | TruQA | Avg. |
|---|---|---|---|---|---|---|---|---|---|---|---|
| | | | | 600M model, 160B tokens from DCLM | | | | | | | |
| Standard | 32.7 | 67.5 | 38.2 | 58.8 | 56.4 | 45.0 | 71.2 | 47.9 | 57.6 | 39.2 | 51.5 |
| MeCo | 32.8 | 67.6 | 37.0 | 62.0 | 54.2 | 47.2 | 71.0 | 49.6 | 57.1 | 37.9 | **51.7** |
| | | | | 1.6B model, 160B tokens from DCLM | | | | | | | |
| Standard | 36.1 | 75.1 | 42.7 | 64.8 | 66.7 | 46.0 | 74.3 | 54.2 | 62.0 | 35.2 | 55.7 |
| MeCo | 36.3 | 75.7 | 44.1 | 63.8 | 67.3 | 51.2 | 73.4 | 52.6 | 64.2 | 38.5 | **56.7** |
| | | | | 3B model, 160B tokens from DCLM | | | | | | | |
| Standard | 39.8 | 76.8 | 48.3 | 66.0 | 74.1 | 49.0 | 76.9 | 56.0 | 66.5 | 38.1 | 59.2 |
| MeCo | 39.7 | 78.6 | 48.5 | 71.0 | 73.6 | 51.8 | 77.0 | 55.5 | 65.9 | 36.4 | **59.8** |
| | | | | 8B model, 80B tokens from DCLM[†] | | | | | | | |
| Standard | 39.2 | 73.3 | 46.0 | 66.0 | 72.8 | 48.8 | 76.1 | 54.8 | 66.2 | 35.2 | 57.8 |
| MeCo | 39.5 | 77.1 | 44.8 | 68.8 | 71.2 | 52.6 | 75.8 | 53.8 | 65.2 | 35.0 | **58.4** |

Table 17: Results with different numbers of parameters. All experiments use the same hyperparameters except for the 8B model[†], which uses a smaller learning rate and fewer tokens due to training instability and limited compute resources.

| Model | MMLU | ARC-e | ARC-c | CSQA | HSwag | OBQA | PIQA | SIQA | WG | TruQA | Avg. |
|---|---|---|---|---|---|---|---|---|---|---|---|
| | | | | 1.6B model, 160B tokens from C4 | | | | | | | |
| Standard | 31.0 | 59.8 | 36.1 | 55.8 | 64.9 | 42.8 | 72.5 | 49.7 | 60.0 | 32.0 | 50.5 |
| MeCo | 31.9 | 62.0 | 37.8 | 54.3 | 63.6 | 43.6 | 74.0 | 50.0 | 58.9 | 39.5 | **51.6** |
| | | | | 1.6B model, 160B tokens from RefinedWeb | | | | | | | |
| Standard | 32.4 | 68.6 | 37.1 | 61.2 | 63.9 | 46.8 | 73.9 | 51.2 | 59.7 | 36.7 | 53.2 |
| MeCo | 32.5 | 69.4 | 38.0 | 61.4 | 64.3 | 48.2 | 73.6 | 53.6 | 60.6 | 38.9 | **54.0** |
| | | | | 1.6B model, 160B tokens from DCLM | | | | | | | |
| Standard | 36.1 | 75.1 | 42.7 | 64.8 | 66.7 | 46.0 | 74.3 | 54.2 | 62.0 | 35.2 | 55.7 |
| MeCo | 36.3 | 75.7 | 44.1 | 63.8 | 67.3 | 51.2 | 73.4 | 52.6 | 64.2 | 38.5 | **56.7** |

Table 18: Detailed results on different pre-training corpora.

| Model | MMLU | ARC-e | ARC-c | CSQA | HSwag | OBQA | PIQA | SIQA | WG | TruQA | Avg. |
|---|---|---|---|---|---|---|---|---|---|---|---|
| | | | | Conditional Inference | | | | | | | |
| Standard | 36.1 | 73.8 | 42.4 | 66.1 | 66.6 | 46.2 | 73.4 | 53.5 | 62.6 | 37.1 | 55.8 |
| MeCo | 36.3 | 74.2 | 44.6 | 65.2 | 67.6 | 51.6 | 73.4 | 53.2 | 66.0 | 40.1 | **57.2** |

Table 19: Full results of using conditional inference (1.6B parameters, 160B tokens).

| Model | MMLU | ARC-e | ARC-c | CSQA | HSwag | OBQA | PIQA | SIQA | WG | TruQA | Avg. |
|---|---|---|---|---|---|---|---|---|---|---|---|
| Standard | 36.1 | 75.1 | 42.7 | 64.8 | 66.7 | 46.0 | 74.3 | 54.2 | 62.0 | 35.2 | 55.7 |
| 100% URL | 33.9 | 72.4 | 28.8 | 37.2 | 61.5 | 42.6 | 72.9 | 52.1 | 60.5 | 41.0 | 50.3 |
| 90% URL + 10% Standard | 36.4 | 72.5 | 43.1 | 63.7 | 66.9 | 50.0 | 75.7 | 53.1 | 62.8 | 39.9 | 56.4 |
| MeCo | 36.3 | 75.7 | 44.1 | 63.8 | 67.3 | 51.2 | 73.4 | 52.6 | 64.2 | 38.5 | **56.7** |

Table 20: Different strategies of mixing metadata-augmented and standard data.

| Model | MMLU | ARC-e | ARC-c | CSQA | HSwag | OBQA | PIQA | SIQA | WG | TruQA | Avg. |
|---|---|---|---|---|---|---|---|---|---|---|---|
| URLs (MeCo) | 36.3 | 75.7 | 44.1 | 63.8 | 67.3 | 51.2 | 73.4 | 52.6 | 64.2 | 38.5 | 56.7 |
| Full URLs | 36.7 | 75.4 | 43.9 | 68.3 | 66.5 | 51.2 | 74.0 | 52.9 | 63.2 | 35.6 | 56.8 |
| URL suffix | 36.2 | 73.9 | 42.7 | 65.2 | 67.7 | 49.0 | 73.1 | 53.6 | 62.1 | 38.1 | 56.2 |
| Top 0.2% URLs | 36.2 | 76.6 | 44.1 | 66.9 | 66.3 | 47.6 | 74.5 | 53.7 | 63.1 | 35.3 | 56.4 |
| Top 2% URLs | 36.5 | 73.5 | 44.8 | 65.4 | 65.8 | 48.2 | 74.3 | 53.4 | 64.3 | 36.9 | 56.3 |
| Hashed URLs | 36.4 | 73.7 | 44.2 | 64.6 | 67.2 | 51.8 | 74.3 | 54.8 | 62.5 | 37.9 | 56.7 |
| Topics | 36.3 | 74.5 | 45.3 | 64.5 | 67.4 | 48.2 | 74.2 | 53.5 | 63.1 | 38.6 | 56.6 |

Table 21: Experiment results on using different types of metadata.

