# OpenReview forum: "Metadata Conditioning Accelerates Language Model Pre-training"
_ICML.cc/2025/Conference — ICML 2025 poster_

### Official Review · Reviewer_mTck · 2025-03-08

**Overall Recommendation:** 2

**Summary:**

The paper proposes a metadata-enhanced training strategy for LLMs across various model sizes, ranging from 600M to 8B parameters. Specifically, during the first 90% of training, metadata is prepended to the training documents, enabling comparable performance while reducing data usage by 33%. The authors conduct experiments on three training corpora to validate the effectiveness of their approach.

**Claims And Evidence:**

1. One of the key claims in this work is that prepending metadata, i.e., URLs, to training documents obtains comparable performance with 33% less training data. As shown on the right-hand side of Figure 1, as the number of training tokens increases from 0 to 80B, 160B, and 240B, the proposed method, Meco, consistently outperforms the baseline. However, this improvement may be influenced by randomness. In other words, even with the same metadata-enhanced training data, different random seeds could impact the average performance curve. Some seeds may lead to rapid performance gains, while others may result in slower improvements. Unfortunately, since this work conducts experiments using only a single seed, the extent to which randomness affects the shape of the average performance curve remains unclear.

2. Additionally, Table 13 only shows the performance of three runs at 160B tokens, it is still unclear how performance change across different training token counts.

**Essential References Not Discussed:**

No

**Experimental Designs Or Analyses:**

1. Experiments are conducted based on a single run, which may weaken the strength of the main claim.
2. Zero-shot evaluation should be included to better assess the effectiveness of the proposed method.

**Methods And Evaluation Criteria:**

1. The primary evaluation in this work is 5-shot, though a very few tasks are evaluated in a zero-shot manner (In Table 3). However, zero-shot evaluation better assesses a model’s ability to generalize, which could provide a stronger indication of the effectiveness of prepending metadata.

**Other Comments Or Suggestions:**

N/A

**Other Strengths And Weaknesses:**

Strengths: the paper is well-structured and easy to follow.
Weaknesses: the paper lacks a theoretical contribution and primarily presents an empirical study on the effect of prepending URLs in LLM training.

**Questions For Authors:**

1. Could you provide results from runs with different random seeds when gradually increasing the number of training tokens?
2. Could you include zero-shot performance for your main experiment results?

**Relation To Broader Scientific Literature:**

Previous studies have explored metadata such as source domains and document IDs, whereas this work leverages URLs, which may be new to the community.

**Theoretical Claims:**

N/A

---

> ### Author Rebuttal · Authors · 2025-03-31
>
> Thank you for your valuable feedback! We address your concerns below:
>
> **Q1: Randomness of the experiment results**
>
> A1: Thank you for raising this point. We acknowledge that a limitation of our study is the lack of multiple runs with different random seeds for most experiments, primarily due to the high cost of pre-training (our main 160B run required 1,536 H100 GPU hours). That said, we would like to emphasize the following:
>
> (1) Single-run experiments are standard in LM pre-training studies, given the resource constraints. Our setup follows established practice, consistent with prior work ([Xie et al., 2023](https://arxiv.org/pdf/2305.10429); [Li et al., 2024](https://arxiv.org/pdf/2406.11794); [Wettig et al., 2024](https://arxiv.org/pdf/2402.09739)). It is also generally accepted that pre-training exhibits less variance than fine-tuning.
>
> (2) As you noted, Table 13 shows low variance of our pre-training experiments (3 different seeds and different subsets of data), particularly when averaging across the full evaluation suite. We believe that the result we present is significant and is not due to randomness.
>
> (3) Figures 3–4 demonstrate consistent gains from MeCo across a range of model sizes and datasets, further supporting that the improvements are not artifacts of randomness but reflect meaningful trends.
>
>
> **Q2: Lack of zero-shot results.**
>
> A2: In this paper, we followed [OLMES](https://arxiv.org/pdf/2406.08446)’s setting, which adopts curated 5-shot examples to reduce evaluation variance. But we agree that adding zero-shot results offers a better picture of model performance. We add a zero-shot evaluation for our main 1.6B, 160B, DCLM experiment as following:
>
> |          | MMLU | ARC-e | ARC-c | CSQA | HSwag | OBQA | PIQA | SIQA | WG   | TruQA | Avg. |
> |----------|------|-------|-------|------|-------|------|------|------|------|--------|------|
> | Standard | 35.1 | 70.7  | 41.4  | 59.5 | 65.3  | 46.6 | 72.9 | 48.9 | 63.8 | 35.6   | 54.0 |
> | MeCo     | 35.5 | 71.0  | 45.4  | 60.6 | 66.2  | 52.4 | 73.0 | 47.3 | 64.9 | 35.8   | 55.2 |
>
> As we can see, the MeCo model still achieves significantly better performance on zero-shot.
>
>
> **Q3: Lack of theoretical contributions**
>
> A3: We acknowledge that this paper did not provide a theoretical justification for MeCo’s effectiveness or a rigorous analysis of how MeCo changes the training dynamics. However, the main contribution of this paper lies in proposing the method and uncovering this interesting phenomenon (metadata conditioning accelerates pre-training), which is both novel and possesses significant empirical impact. Theoretical understanding of such a method is challenging due to the nature of pre-training, and is also beyond the scope of this empirical study.
>
> That said, we included empirical ablations and hypotheses to shed light on the possible inner-workings of MeCo. In Sec 5.2 and Table 5, we showed that using hashed URLs can achieve similar performance as natural URLs, suggesting that the semantic meaning of the URLs is not necessary for better pre-trained models, and MeCo mostly provides signals to group similar documents together. We agree with reviewer TsUA that with the grouping information, the models can either learn to upweight certain “higher-quality” domains (such as Wikipedia) or learn an implicit curriculum that helps accelerate training—but it is still unclear to us how exactly MeCo changes the training, and it warrants further investigation.
>
> We also highlight a recent preprint, [Zhu et al., On the Power of Context-Enhanced Learning in LLMs](https://arxiv.org/pdf/2503.01821). Though their setting is synthetic and differs from ours, their theoretical analysis shows that context-enhanced learning—such as providing metadata at the beginning of the sequence without actually optimizing cross entropy loss on these tokens—can improve sample efficiency. We found the result insightful and encourage the reviewers to check it out as well.

---

### Official Review · Reviewer_HiPa · 2025-03-09

**Overall Recommendation:** 3

**Summary:**

This paper proposes to include metadata (source links) in the pre-training of language models to boost learning efficiency. The proposed method, MeCo, pre-trains language models with text augmented with metadata in the first 90% of data and the metadata are removed in the last 10% of data for “cooldown”. MeCo is benchmarked on various commonsense reasoning datasets to show better performance and is also more steerable to different generation styles conditioned on different metadata.

**Claims And Evidence:**

1. MeCo is claimed to be a more efficient pre-training paradigm. However, the included benchmarks only involve commonsense reasoning, which fails to comprehend all kinds of abilities in LMs.
2. MeCo is claimed to make LM more steerable, which is well-validated by various ablation studies and harmfulness reduction experiments.

**Essential References Not Discussed:**

The related literature is well discussed.

**Experimental Designs Or Analyses:**

The experiment design is reasonable.

**Methods And Evaluation Criteria:**

The benchmarks are mostly commonsense reasoning, which is a bit narrow to support the claim that MeCo is universally more efficient. The biggest concern is there to be various sources for pre-training as shown in Table 15. However, commonsense reasoning can only benchmark a few datasets (e.g. Wikipedia) in the corpus.

**Other Comments Or Suggestions:**

This paper proposes an interesting idea, which is worth further discussion. However, it's not ready for publication because its narrow scope of benchmarking, explanation of the pre-training efficiency, and limitation in the case that all metadata are available.

**Other Strengths And Weaknesses:**

Another weakness of MeCo is the requirement for metadata to be appended to the raw text. The setup in this paper discusses only the case that metadata are available to all pre-training data. However, most curated raw texts do not contain their metadata. Then, will the steerability observed from pre-training with 90% w/ metadata + 10% w/o metadata still appear in the case that most data are without metadata? I feel the conclusion in this paper might not be applicable to larger-scale pre-training, which requires further discussion.

**Questions For Authors:**

It is intuitive that metadata enables more steerable language, and may also make sense for better pre-training efficiency. But it's not intuitive that the resulting model will perform better. The improvement is also only shown in commonsense reasoning. Can you provide some intuition as to why metadata can improve LM's performance?

**Relation To Broader Scientific Literature:**

This paper is connected to the source for pre-training language models, this paper incorporates metadata into the pre-training to enable better generation steering.

**Theoretical Claims:**

N/A, The paper is more about an empirical claim.

---

> ### Author Rebuttal · Authors · 2025-03-31
>
> Thank you for your valuable feedback! We address your concerns here:
>
> **W1: Evaluation only includes commonsense reasoning and only reflects a few sources such as Wikipedia.**
>
> A1: **We evaluate our models by using OLMES ([Gu et al., 2024](https://arxiv.org/abs/2406.08446v1)), the industry-standard evaluation behind AI2’s OLMo models ([Groeneveld et al., 2024](https://arxiv.org/abs/2402.00838v3))**. Similar tasks are also used in [Llama](https://arxiv.org/abs/2302.13971) and [Open LLM Leaderboard](https://huggingface.co/spaces/open-llm-leaderboard-old/open_llm_leaderboard). While our evaluation does not include some of the latest popular LLM benchmarks on math (GSM8K, MATH), coding (HumanEval), graduate-level QA (GPQA), or instruction following (AlpacaEval, IFEval), we emphasize that at our experiment scale (1B model, 160B tokens), models do not achieve any meaningful performance on math, coding, and graduate-level QA—making these benchmarks uninformative. Additionally, since our study focuses on pre-training instead of post-training, the models are not expected to perform well on instruction-following tasks. Other studies that similarly investigate pre-training (e.g., [Wettig et al., 2024](https://arxiv.org/pdf/2402.09739); [Yu et al., 2024](https://arxiv.org/pdf/2406.06046)) adopt comparable evaluation setups.
>
> **We also stress that OLMES covers more than just “commonsense”**. For example, MMLU benchmarks model knowledge on a diverse set of subjects, including math, medicine, and economics; OpenbookQA and TruthfulQA test models’ factual correctness. This suite of tasks has been used by numerous pre-training studies and industry LLMs, and are considered to cover a diverse range of capabilities and to picture a holistic picture of the model performance. **We respectfully disagree with the claim that the evaluation reflects only a narrow set of sources such as Wikipedia**—for example, websites like personal blog-posts often benefit models’ commonsense performance.
>
>
> **W2:  MeCo only discussed the case where all metadata is available. Most curated raw texts do not contain metadata.**
>
> A2: We respectfully disagree with the reviewer on “most curated raw texts do not contain metadata”. Most pre-training sources, such as CommonCrawl, C4, FineWeb, RefinedWeb, and DCLM, provide at least the URL information. For companies that perform their own data crawling, retaining metadata like URLs is standard practice.
>
>
> **W3: Explanation of the pre-training efficiency**
>
> A3: Thanks for raising this point! We included empirical ablations and hypotheses to shed light on the possible inner-workings of MeCo. In Sec 5.2 and Table 5, we showed that using hashed URLs can achieve similar performance as natural URLs, suggesting that the semantic meaning of the URLs is not necessary for better pre-trained models, and MeCo mostly provides signals to group similar documents together. We agree with reviewer TsUA that with the grouping information, the models can either learn to upweight certain “higher-quality” domains (such as Wikipedia) or learn an implicit curriculum that helps accelerate training—but it is still unclear to us how exactly MeCo changes the training, and it warrants further investigation.
>
> We also highlight a recent preprint, [Zhu et al., On the Power of Context-Enhanced Learning in LLMs](https://arxiv.org/pdf/2503.01821). Though their setting is synthetic and differs from ours, their theoretical analysis shows that context-enhanced learning—such as providing metadata at the beginning of the sequence without actually optimizing cross entropy loss on these tokens—can improve sample efficiency. We found the result insightful and encourage the reviewers to check it out as well.

---

### Official Review · Reviewer_kHX6 · 2025-03-13

**Overall Recommendation:** 4

**Summary:**

The paper presents a novel method named Metadata Conditioning then Cooldown (MeCo), which appends metadata (primarily URLs) to pretraining documents and significantly accelerates pre-training. The authors also show how MeCo can be used for model steering by conditioning prompts on metadata, enhancing both downstream task performance and reducing harmful outputs. This approach is working probably because the model can perform data grouping based on data sources.

**Claims And Evidence:**

The claims are well supported by experiments. The main claims include:
1. Accelerated Pre-training.
2. Improved Downstream.
3. Model Steerability.
4. Metadata can have different types.
5. Minimal extra computational cost.

**Essential References Not Discussed:**

I don't have comments on this section.

**Experimental Designs Or Analyses:**

The Experimental Designs are straightforward and intuitive. They can support the claims of the paper.

**Methods And Evaluation Criteria:**

The Methods And Evaluation are properly performed, including comparing PPL and downstream tasks scores.

**Other Comments Or Suggestions:**

No other comments.

**Other Strengths And Weaknesses:**

Stengths:
1. Extensive evaluation across multiple tasks, datasets, and model sizes.
2. Insightful ablation studies.
3. Clear, practical and easy method.
4. The most interesting thing to me is that using hashed URLs can also do the trick.

Weaknesses:
1. Lack of Theoretical Explanation: Empirical evidence is provided, but I am not really sure how it changes the trianing dynamics.
2. The choice of Cooldown Strategy seems effective, but arbitrary. The paper does not thoroughly justify the chosen duration.
3. Although MeCo was tested across several standard benchmarks and datasets, the paper did not extensively evaluate its generalization to languages beyond English or explicitly measure robustness across more diverse downstream tasks.

**Questions For Authors:**

N/A

**Relation To Broader Scientific Literature:**

The method can greatly accelerates pre-training and enhances model steerability without introducing computational overhead or limited applicability. It can be very helpful to the pretraining domain.

**Theoretical Claims:**

There are not too much theoretical claims. Most of the claims are emprical and based on experiment results, but I think they make sense intuitively.

---

> ### Author Rebuttal · Authors · 2025-03-31
>
> Thank you for your suggestions and questions! We appreciate that you recognize the paper’s contributions and strengths. To address your concerns:
>
> **W1: Lack of theoretical explanation—how does MeCo change the training dynamics?**
>
> A1: Thank you for raising this point! We acknowledge that this paper did not provide a theoretical justification for MeCo’s effectiveness or a rigorous analysis of how MeCo changes the training dynamics. However, the main contribution of this paper lies in proposing the method and uncovering this interesting phenomenon (metadata conditioning accelerates pre-training), which is both novel and possesses significant empirical impact. Theoretical understanding of such a method is challenging due to the nature of pre-training, and is also beyond the scope of this empirical study.
>
> That said, we included empirical ablations and hypotheses to shed light on the possible inner-workings of MeCo. In Sec 5.2 and Table 5, we showed that using hashed URLs can achieve similar performance as natural URLs, suggesting that the semantic meaning of the URLs is not necessary for better pre-trained models, and MeCo mostly provides signals to group similar documents together. We agree with reviewer TsUA that with the grouping information, the models can either learn to upweight certain “higher-quality” domains (such as Wikipedia) or learn an implicit curriculum that helps accelerate training—but it is still unclear to us how exactly MeCo changes the training, and it warrants further investigation.
>
> We also highlight a recent preprint, [Zhu et al., On the Power of Context-Enhanced Learning in LLMs](https://arxiv.org/pdf/2503.01821). Though their setting is synthetic and differs from ours, their theoretical analysis shows that context-enhanced learning—such as providing metadata at the beginning of the sequence without actually optimizing cross entropy loss on these tokens—can improve sample efficiency. We found the result insightful and encourage the reviewers to check it out as well.
>
>
> **W2: Arbitrary duration choice for cooldown**
>
> A2: We provide an ablation study on the duration of cooldown in Table 14 (Appendix B.3), which demonstrates that 10% cooldown leads to competitive performance.
>
> **W3: Though the authors conducted evaluation on some standard benchmarks, there lacks (1) non-English tasks and (2) evaluation on robustness across more diverse tasks.**
>
> A3: Thanks for raising this point!
> We evaluate our models by using OLMES ([Gu et al., 2024](https://arxiv.org/abs/2406.08446v1)), the industry-standard evaluation behind AI2’s OLMo models ([Groeneveld et al., 2024](https://arxiv.org/abs/2402.00838v3)). Similar tasks are also used in [Llama](https://arxiv.org/abs/2302.13971) and [Open LLM Leaderboard](https://huggingface.co/spaces/open-llm-leaderboard-old/open_llm_leaderboard). While our evaluation does not include some of the latest popular LLM benchmarks on math (GSM8K, MATH), coding (HumanEval), graduate-level QA (GPQA), or instruction following (AlpacaEval, IFEval), we emphasize that at our experiment scale (1B model, 160B tokens), models do not achieve any meaningful performance on math, coding, and graduate-level QA—making these benchmarks uninformative. Additionally, since our study focuses on pre-training instead of post-training, the models are not expected to perform well on instruction-following tasks. Other studies that similarly investigate pre-training (e.g., [Wettig et al., 2024](https://arxiv.org/pdf/2402.09739); [Yu et al., 2024](https://arxiv.org/pdf/2406.06046)) adopt comparable evaluation setups.
>
> That said, we acknowledge that the evaluation suite we use focuses on English tasks, which is a common limitation in language model pre-training studies.

---

### Official Review · Reviewer_TsUA · 2025-03-14

**Overall Recommendation:** 5

**Summary:**

This paper proposes Metadata Conditioning then Cooldown (MeCo) to accelerate LM pre-training. MeCo starts with pre-training LMs with metadata (URL's absolute domain) prepended in front of the text in the first 90% of pre-training and uses text only (no metadata) for pre-training in the last 10% of pre-training. They conduct experiments with decoder-only LMs of four scales on three pre-training data and show that MeCo improves the performance on most downstream tasks and obtains an average performance gain of 1.0. They conduct extensive ablations to justify the design choice in MeCo and provides partial explanation for its success.

**Claims And Evidence:**

The main claim is "MeCo accelerates LM pre-training". The paper provides sufficient and clear evidence for this, with LM pre-trained on three pre-training datasets and on four model scales. The average performance improvement is convincing.

**Essential References Not Discussed:**

N/A

**Experimental Designs Or Analyses:**

The experiment design is very sound

**Methods And Evaluation Criteria:**

The evaluation is good. Using OLMES, an evaluation suite from Ai2, for evaluation, makes the evaluation convincing and reproducible.

**Other Comments Or Suggestions:**

I like this paper a lot

**Other Strengths And Weaknesses:**

Strengths
===
- The paper is very well-written. I enjoy how the authors arrange the contents and cross-reference results in the latter part of the paper to support their claims in Section 1 and Section 2.
- The method, MeCo, is simple, elegant, and effective.
- The evaluation is reproducible.
- The related works are properly discussed

Weaknesses
===
None

**Questions For Authors:**

- Q1. There is a paper [1], not strictly relevant but somewhat related, that shows that LLM's answer does not seem to be affected when changing the metadata of the retrieved documents in RAG. This somewhat contradicts the results shown in this paper, showing that adding proper metadata helps, and adversarial metadata deteriorates the results. It would be interesting to hear the author's comments on this and how MeCo may or may not be better at distinguishing more reliable sources in a retrieval-augmented generation. It will also be beneficial to include this discussion in the paper. However, this is not a requirement for the paper.

- Q2. I am personally very interested in understanding why MeCo is more efficient, and I believe that the research community will also be interested in this. While the paper does provide some explanations, I think they are far from complete, as acknowledged by the authors in Section 5.2. I think studying pre-training dynamics may reveal some interesting observations about MeCo. For example, does MeCo prioritize learning the data from certain domains? Does MeCo's training automatically enable some curriculum for the LM? It would be beneficial if the authors could also release the intermediate checkpoints of these models, as pre-training is computationally infeasible for most research groups.



- [1] [Do Metadata and Appearance of the Retrieved Webpages Affect LLM’s Reasoning in Retrieval-Augmented Generation?](https://aclanthology.org/2024.blackboxnlp-1.24/) (Chiang & Lee, BlackboxNLP 2024)

**Relation To Broader Scientific Literature:**

This paper shows that conditioning on metadata improves pre-training efficiency. While prior works do use metadata or some tags to steer LM generations, using metadata to speed up pre-training has never been explored. This is a novel and important contribution to the community.

**Theoretical Claims:**

N/A

---

> ### Author Rebuttal · Authors · 2025-03-31
>
> Thanks for your positive review! We are glad that you found our method interesting and our experiment design sound. To answer your questions:
>
> **Q1: Chiang and Lee, 2024 show that providing different source information often does not affect RAG results. How to interpret this?**
>
> A1: Thanks for providing the reference and we’ll include a discussion of this work. We believe the influence of metadata depends on both the metadata type and the model’s training. As noted by Chiang and Lee, publication time significantly impacts RAG performance, while source does not—indicating higher sensitivity to temporal information. However, under MeCo training, models also become sensitive to source metadata as it’s explicitly conditioned on such information—this is evidenced by our conditional inference result.
>
>
> **Q2: Why does MeCo work? Can you release intermediate checkpoints?**
>
> A2: Thanks for raising this point! We included empirical ablations and hypotheses to shed light on the possible inner-workings of MeCo. In Sec 5.2 and Table 5, we showed that using hashed URLs can achieve similar performance as natural URLs, suggesting that the semantic meaning of the URLs is not necessary for better pre-trained models, and MeCo mostly provides signals to group similar documents together. We agree with the reviewer that with the grouping information, the models can either learn to upweight certain “higher-quality” domains (such as Wikipedia) or learn an implicit curriculum that helps accelerate training—but it is still unclear to us how exactly MeCo changes the training, and it warrants further investigation.
>
> We will also make sure to release all intermediate checkpoints to facilitate future research.

---

> > ### Comment · Reviewer_TsUA · 2025-04-02
> >
> > Thank you for your response and for agreeing to share the intermediate checkpoints. I keep my original evaluation. This is a good paper that should be accepted.

---

### Decision · Program_Chairs · 2025-05-01

**Decision:**

Accept (poster)

**Comment:**

The submission got one Strong Accept, one Accept, one Weak Accept, and one Weak Reject. After reading the reviews and discussion, most reviewers maintained their scores and sound positive about the submission. The paper proposes MeCo, a method to accelerate LM pre-training. Overall, reviewers think the method is well executed, but lacks theoretical analysis. Nevertheless, I think this method is of interest to the community and I think it can have practical benefits and trigger interest in the community. I would encourage the authors to include a discussion about limitations and additional experiments (especially the zero-shot results) in a final revision.For these reasons, I am recommending acceptance.